# Diagnostic accuracy of SOX11 immunohistochemistry in mantle cell lymphoma: A meta-analysis

**Woojoo Lee**[1☯]**, Eun Shin**[2☯]**, Bo-Hyung Kim**[3,4]**, Hyunchul Kim**[2]***

**1** Department of Statistics, Inha University, Incheon, Republic of Korea, **2** Department of Pathology, Hallym University Dongtan Sacred Heart Hospital, Hwaseong-si, Gyeonggi-do, Republic of Korea, **3** Department of Clinical Pharmacology and Therapeutics, Kyung Hee University College of Medicine and Hospital, Seoul, Republic of Korea, **4** Department of Biomedical Science and Technology, Graduate School, Kyung Hee University, Seoul, Republic of Korea

☯ These authors contributed equally to this work.
* hyunchulk@hallym.or.kr

**Data Availability Statement:** All relevant data are within the manuscript and its Supporting Information files.

**Funding:** This work was supported by a Grant from the Next-Generation BioGreen 21 program (Project

## Abstract

SOX11 is a transcription factor that is normally expressed in the fetal brain and has also been detected in some malignant tumors, including mantle cell lymphoma (MCL). MCL is a mature B-cell lymphoma that characteristically expresses cyclin D1, which has been used as a diagnostic tumor marker. SOX11 has also recently emerged as a tumor marker for MCL, particularly in cyclin D1-negative MCLs and to distinguish between MCLs and other cyclin D1-positive lymphomas. In this study, we evaluated the diagnostic accuracy of SOX11 immunohistochemistry for the diagnosis of MCL using a meta-analysis. A comprehensive literature search was performed using the PubMED, EMBASE, and Cochrane library through May 9, 2018. In total, 14 studies were included in our meta-analysis. The sensitivity, specificity, and area under the curve calculated from the summary receiver operator characteristic were 0.9, 0.95, and 0.934, respectively. Effect sizes of log positive likelihood ratios, log negative likelihood ratios, and log diagnostic odds ratios were 2.67, -2.12, and 5.27, respectively. Statistically significant substantial heterogeneity was observed for specificity ($I^2 = 95\%$), but not for sensitivity. Subgroup analysis and meta-regression were performed to explain the heterogeneity in specificity and showed that the proportions of Burkitt's lymphoma, lymphoblastic lymphoma, and hairy cell leukemia were significant covariates among studies using rabbit polyclonal antibodies. Overall, this meta-analysis showed that SOX11 was a useful diagnostic marker for MCL, with the clone MRQ-58 mouse monoclonal antibody showing particularly robust performance.

## Introduction

SOX11 is a transcription factor that is normally expressed in the fetal brain and is thought to play a role in nervous system development.[1] SOX11 is also expressed in several neoplastic conditions, including ovarian carcinomas, pancreatic solid pseudo-papillary tumor, brain

No. PJ01337701), Rural Development Administration, Republic of Korea to WL. The funder had no role in study design, data collection and analysis, decision to publish, or preparation of the manuscript.

**Competing interests:** The authors have declared that no competing interests exist.

tumors, and lymphomas.[2–5] Among lymphomas, mantle cell lymphoma (MCL) shows higher expression of SOX11 than other types of lymphoproliferative disorders (LPDs).[6]

MCL is a mature B-cell lymphoma characterized by expression of CD5 and cyclin D1.[7] Cyclin D1 expression is a result of t(11;14)(q13:q32) translocation between the *IGH* gene and the *CCND1* gene.[8, 9] However, the diagnosis of MCLs can be complicated in some cases. For example, cyclin D1-negative MCLs[10] can overexpress cyclin D2 or cyclin D3,[11] and aggressive MCLs must be distinguished from cyclin D1-positive diffuse large B cells.[12] In such cases, SOX11 has emerged as a potential novel diagnostic marker of MCL.[6, 13]

As a diagnostic marker of MCL, the overall diagnostic accuracy of SOX11 in MCL has not yet been evaluated. Additionally, a comprehensive analysis of its reliability issues, including its low specificity[5] and high false-positive rates in Burkitt's lymphoma (BL), lymphoblastic lymphoma (LBL), and hairy cell leukemia (HCL),[5, 14, 15] has not been performed.

Accordingly, in this meta-analysis, we evaluated the diagnostic accuracy of SOX11 immunohistochemistry for MCL. Additionally, we assessed the cause of the inconsistent specificity by comparing the specificities of different antibody clonalities and different monoclonal antibodies using subgroup analysis. Finally, meta-regression was carried out to determine the proportions of BL, LBL, and HCL, which could affect the specificity of SOX11 across different antibodies.

## Materials and methods

### Published studies and selection criteria

We searched PubMed, EMBASE, and Cochrane library through May 9, 2018 with the following key words: "SOX11" and ("lymphoma" or "lymphomas"). Reference lists of review articles were also searched. Duplicate data and articles were excluded considering the authors and their affiliations. Original articles were included if SOX11 immunohistochemistry was performed in human MCL and other LPD cases. When multiple articles from an author or institution were found, the most informative article was selected for the current study. Non-English articles, article or conference abstracts without sufficient information for meta-analysis, review articles, case reports, comments, errata, articles on cell lines or animals, articles with SOX11 immunohistochemistry on MCL only without other LPD, and those concerning SOX11 studies with methods other than IHC were excluded. The selection process is shown in Fig 1.

### Data extraction

The following data from all eligible studies were extracted[5, 6, 12–23]: the first author's name, year of publication, species and clonality of the anti-SOX11 antibody, clone or catalog number of the anti-SOX11 antibody, number of SOX11-positive MCLs (true positive [TP]), number of total MCLs (number of cases), number of SOX11-positive other LPDs (false positives), number of total other LPDs (number of controls), sensitivity, specificity, and numbers of SOX11-positive and total BL, LBL, and HCL (BL+LBL+HCL positive/total).

The Quality Assessment of Diagnostic Accuracy Studies (QUADAS) tool was applied for quality assessment of each study.[24] QUADAS consists of 14 questions, which are scored yes (score = 1), no (score = 0), or unclear (score = 0).

### Statistical analyses

All data were analyzed using R version 3.4.3, with the "meta" and "mada" packages.[25–27] We calculated the sensitivity and specificity, and the results were visualized on Forest plots

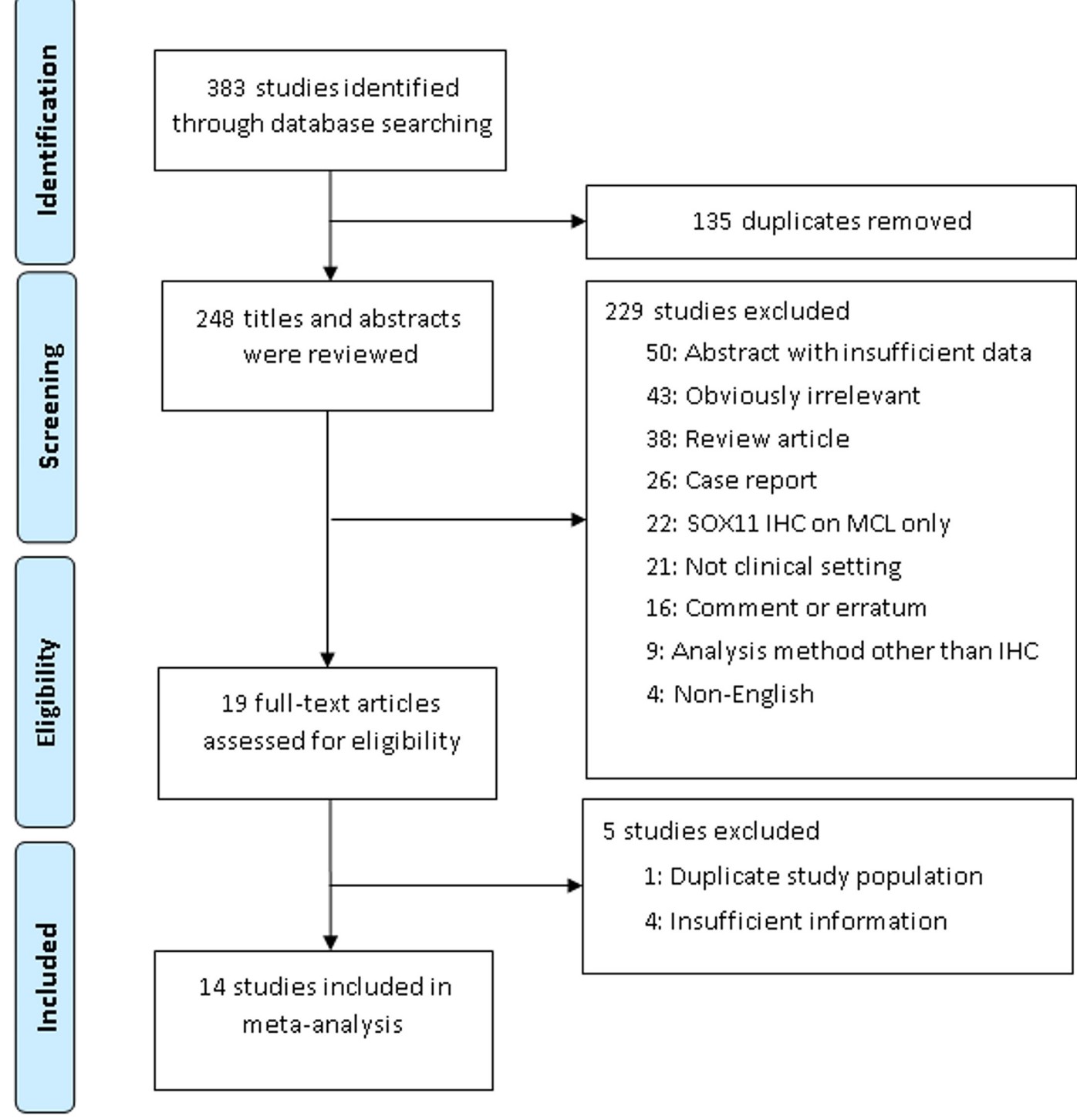

**Fig 1. Flow diagram of study selection.**

with 95% confidence intervals (CIs). Based on random effect models, statistical heterogeneity was evaluated using Higgins' $I^2$ statistics. In our meta-analysis, studies with $I^2$ values of greater than 50% were considered substantially heterogeneous. The sensitivity and specificity of each study were used to plot the summary receiver operating characteristic (SROC) curve and

calculate the area under the SROC curve (AUC). Publication bias was examined by the test for funnel plot asymmetry based on a linear regression model.[28] Subgroup analysis was performed for specificity by setting the species and clonality of the antibodies and clone of the monoclonal antibodies as moderators. Meta-regression analyses were performed for specificity with proportions of BL, LBL, and HCL among other LPDs (control) as covariates in all studies for mouse monoclonal antibodies and rabbit polyclonal antibodies. Residual heterogeneity, which could not be explained by the covariate used in the meta-regression, was also considered present when $I^2$ values were greater than 50%. Results with $P$ values of less than 0.05 were considered as statistically significant.

## Results

### Characteristics of the studies

Three hundred eighty-three reports were identified in the database search. In total, 14 studies fulfilled the inclusion criteria[5, 6, 12–23]; all were case-control studies. Two studies used more than one antibody [19, 21]. Rabbit polyclonal antibodies were used for seven study populations[5, 6, 14–17, 22]; mouse monoclonal antibodies were used for eight study populations [12, 18–21, 23]. A goat polyclonal antibody was used for one study population.[21] One study did not specify the species of antibody used.[13] Among the studies with mouse monoclonal antibodies, clone MRQ-58 was used in five study populations.[12, 18, 19, 21, 23] BL, LBL, and HCL were included in 12 study populations.[5, 6, 14, 15, 17, 19–21, 23] The proportions of LB, LBL, and HCL among other LPD cases ranged from 0.06 to 37 (Table 1). For all studies, meta-analysis was performed using random effect models. Quality assessment based on QUADAS guidelines was conducted for the included studies (S1 Table). The QUADAS scores of the studies ranged from 6 to 11 (Table 1).

### Diagnostic accuracy of SOX11 for MCL

The Forest plots of sensitivity, specificity, log positive likely hood ratio (PLR), log negative likelihood ratio (NLR), and log diagnostic odds ratio (DOR) are shown in Fig 2. The sensitivity of SOX11 for the diagnosis of MCL ranged from 78% to 100%, and the specificity ranged from 72% to 100%. The $I^2$ values of sensitivity and specificity were 49% ($p = 0.01$) and 95% ($p < 0.01$), respectively, indicating that substantial heterogeneity existed in specificity among the eligible studies. The sensitivity and specificity of the studies were plotted in a SROC curve (Fig 3). The sensitivity and specificity (1-False positive rate) calculated from the hierarchical SROC were 0.9 (95% CI, 0.86–0.92) and 0.95 (95% CI, 0.9–0.97). The AUC of SROC was 0.934, indicating that SOX11 may be able to differentiate MCLs from other LPDs with relatively high accuracy. The Spearman correlation coefficient between the logit of sensitivity and 1-speicificty was 0.099 ($p = 0.7048$), suggesting that there was no significant threshold effect. The effect sizes of log PLR, log NLR, and log DOR were 2.67 (95% CI, 2.18–3.17), -2.12 (95% CI, -2.45–-1.78), and 5.27 (95% CI, 4.40–6.14), respectively. A good diagnostic test should have large log PLR, small log NLR, and large DOR. The overall effect sizes of log PLR and log NLR, which are significantly different from 0 ($p < 0.05$), imply that SOX11 immunohistochemistry has diagnostic value for MCL. Following McGee,[29] the overall effect size of DOR also showed clear evidence of SOX11 immunohistochemistry as a diagnostic test for MCL.

### Publication bias

We used Thompson and Sharp's test for funnel plot asymmetry to check whether there was evidence of publication bias in our collection of studies.[28] The funnel plot and test did not

**Table 1.  Characteristics of studies reporting SOX11 immunohistochemistry in mantle cell lymphoma and other lymphoproliferative diseases.**

| Study | Ab species and clonality | Clone (Cat. Number) | Case/Controls | TP | FP | FN | TN | BL+LBL+HCL positive/total | Proportion of BL+LBL+HCL total/controls | QUADAS |
|---|---|---|---|---|---|---|---|---|---|---|
| 2008 Wang | rabbit poly | (HPA000536)* | 53/12 | 48 | 0 | 5 | 12 | None | 0 | 8 |
| 2009 Dictor | rabbit poly | self-made | 23/149 | 18 | 31 | 5 | 118 | 31/45 | 0.3 | 11 |
| 2009 Mozos | rabbit poly | (HPA000536)* | 66/209 | 62 | 11 | 4 | 198 | 8/14 | 0.07 | 9 |
| 2010 Chen | rabbit poly | (HPA000536)* | 57/154 | 54 | 5 | 3 | 149 | 5/10 | 0.06 | 8 |
| 2012 Cao | rabbit poly | (sc-20096) | 4/11 | 3 | 0 | 1 | 11 | None | 0 | 7 |
| 2012 Hsiao | N/A | N/A | 19/98 | 17 | 0 | 2 | 98 | None | 0 | 7 |
| 2012 Nordstrom | mouse mono | self-made | 16/46 | 15 | 2 | 1 | 44 | 2/17 | 0.37 | 6 |
| 2012 Zeng | rabbit poly | N/A | 35/110 | 35 | 5 | 0 | 105 | 5/10 | 0.09 | 9 |
| 2013 Zhang | rabbit poly | N/A | 58/291 | 54 | 81 | 4 | 210 | 16/63 | 0.22 | 9 |
| 2014 Nakashima MRQ58 | mouse mono | MRQ-58 | 80/134 | 77 | 5 | 3 | 129 | 3/14 | 0.1 | 9 |
| 2014 Nakashima CL0142 | mouse mono | CL0142 | 41/95 | 41 | 26 | 0 | 69 | 7/10 | 0.11 | 9 |
| 2014 Soldini MRQ58 | mouse mono | MRQ-58 | 32/173 | 29 | 0 | 3 | 173 | 0/40 | 0.23 | 10 |
| 2014 Soldini CL0143 | mouse mono | CL0143 | 36/173 | 29 | 31 | 7 | 142 | 28/40 | 0.23 | 10 |
| 2014 Soldini sc-17347 | Goat poly | (sc-17347) | 32/145 | 27 | 10 | 5 | 135 | 7/39 | 0.27 | 10 |
| 2014 Zhang | mouse mono | MRQ-58 | 13/46 | 13 | 0 | 0 | 46 | 0/9 | 0.2 | 7 |
| 2016 Hsi | mouse mono | MRQ-58* | 8/63 | 7 | 0 | 1 | 63 | None | 0 | 8 |
| 2017 Chuang | mouse mono | MRQ-58 | 10/490 | 9 | 1 | 1 | 489 | None | 0 | 8 |

Ab: antibody; Rabbit Poly: rabbit polyclonal; Mouse Mono: mouse monoclonal; Goat poly: goat polyclonal; N/A: not available; Cat. number: catalog number; TP: true positive; FP: false positive; FN: false negative; TN: true negative; BL: Burkitt's lymphoma, LBL: Lymphoblastic lymphoma, HCL: Hairy cell leukemia

*: the data was not specified on the study and retrieved from vender's homepage

show significant results at the 0.05 level ($p = 0.136$ for sensitivity and $p = 0.420$ for specificity; Fig 4A and 4B).

## Subgroup analysis

Because substantial heterogeneity was present in the overall analysis of specificity, subgroup analysis was performed to explore heterogeneity further. Due to the limited number of studies, a univariate approach was employed.

The first categorical variable was the antibody clonality. All studies, except one without specified antibody species and clonality, were grouped according to their clonality, i.e., goat polyclonal, rabbit polyclonal, and mouse monoclonal[5, 6, 12, 14–23]; subgroup analysis was performed in an attempt to explain the source of heterogeneity. High levels of within group and between group heterogeneity were present (Fig 5A). The results indicated that the specific clonality of the antibody could not explain the inconsistency in specificity.

The second categorical variable was the clone of monoclonal antibodies. Studies with monoclonal antibodies were divided into two groups: those with clone MRQ-58 and others, [12, 18–21, 23] and subgroup analysis was performed to determine whether the source of heterogeneity could be explained by the monoclonal antibody clone. Both groups showed high heterogeneity. The clone MRQ-58, which was expected to show a more homogeneous result, actually showed less heterogeneity ($I^2 = 62\%$, $p = 0.03$) than the other antibodies ($I^2 = 85\%$, $p < 0.01$; Fig 5B). However, the specificity of the studies in the MRQ-58 group was 1 or near 1, indicating that although the specificity of the MRQ-58 group was statistically heterogeneous, this antibody could be clinically regarded as highly specific.

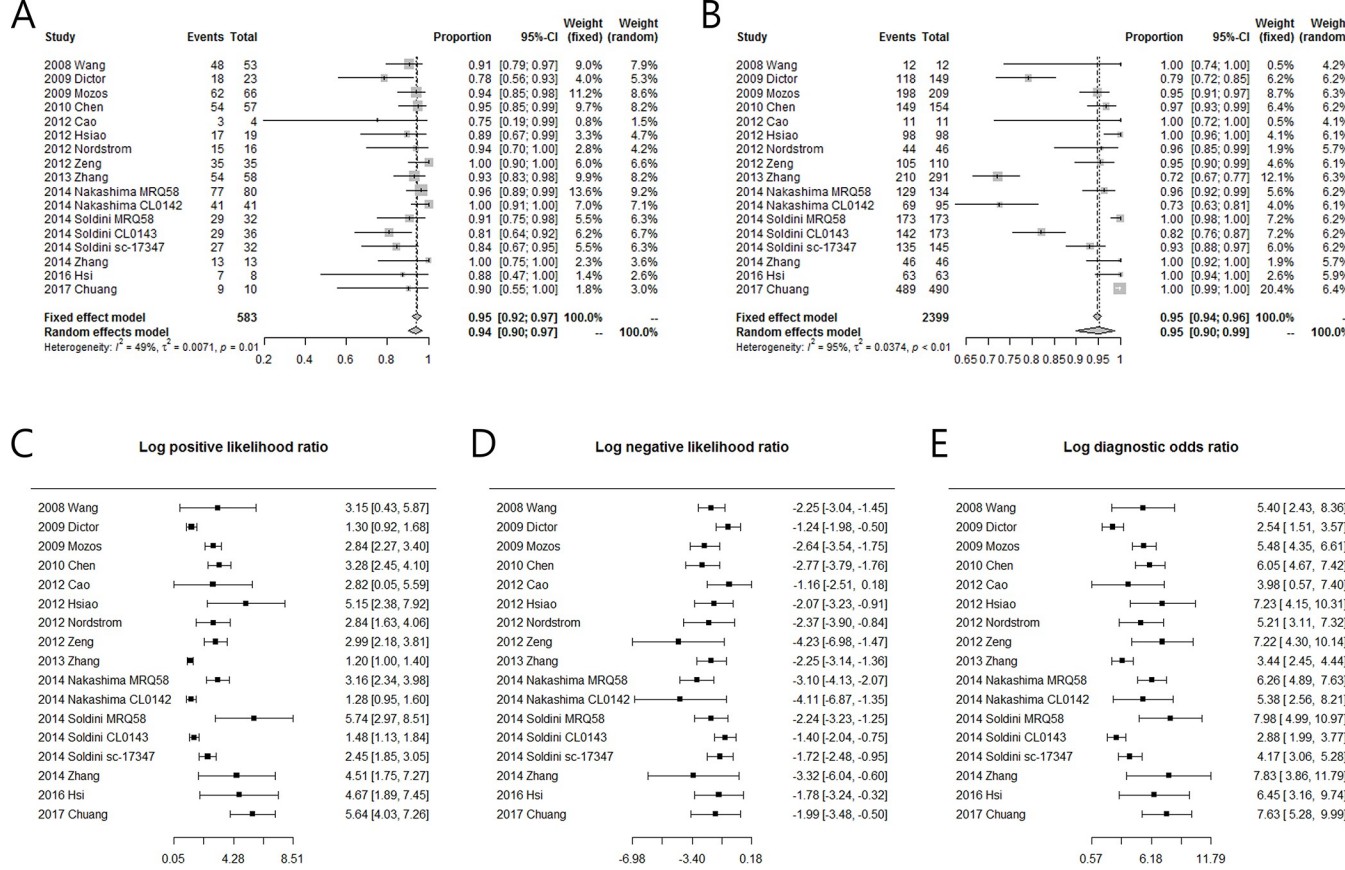

**Fig 2. Forest plot for measures of diagnostic accuracy of SOX11 immunohistochemistry as a diagnostic marker for mantle cell lymphoma.** (A) sensitivity, (B) specificity, (C) log PLR, (D) log NLR, and (E) log DOR.

## Meta-regression

Because false-positive cases were frequent among BL, LBL, and HCL in the included studies, meta-regression was performed to further explore the effects of the proportions of BL, LBL, and HCL in controls to heterogeneity among studies. Meta-regression was performed for three different study groups: all studies, studies using mouse monoclonal antibodies, and studies using rabbit polyclonal antibodies with proportions of BL, LBL, and HCL in controls as a covariate. Due to the limited number of studies, a univariate approach was employed (Table 2).

First, meta-regression was performed for all studies with specified cases of BL, LBL, and HCL.[5, 6, 14, 15, 17, 19–21, 23] The results showed that the proportions of BL, LBL, and HCL were not statistically significant covariates among all studies (intercept = 1.32, 95% CI, 1.09–1.55, $p < 0.0001$; slope = -0.23, 95% CI, -1.33–-0.86, $p = 0.68$). The meta-regression showed substantial residual heterogeneity ($I^2 = 94.8\%$), indicating that the proportions of BL, LBL, and HCL did not influence the specificity of all studies and that substantial residual heterogeneity was present in specificity after considering the effects of BL, LBL, and HCL.

Second, meta-regression was performed on studies using mouse monoclonal antibodies with specified cases of BL, LBL, and HCL.[19–21, 23] The results showed that the proportions of BL, LBL, and HCL were not significant covariates among studies using mouse monoclonal antibodies (intercept = 1.2, 95% CI, 0.73–1.67, $p < 0.0001$; slope = 0.56, 95% CI, -1.56–2.64,

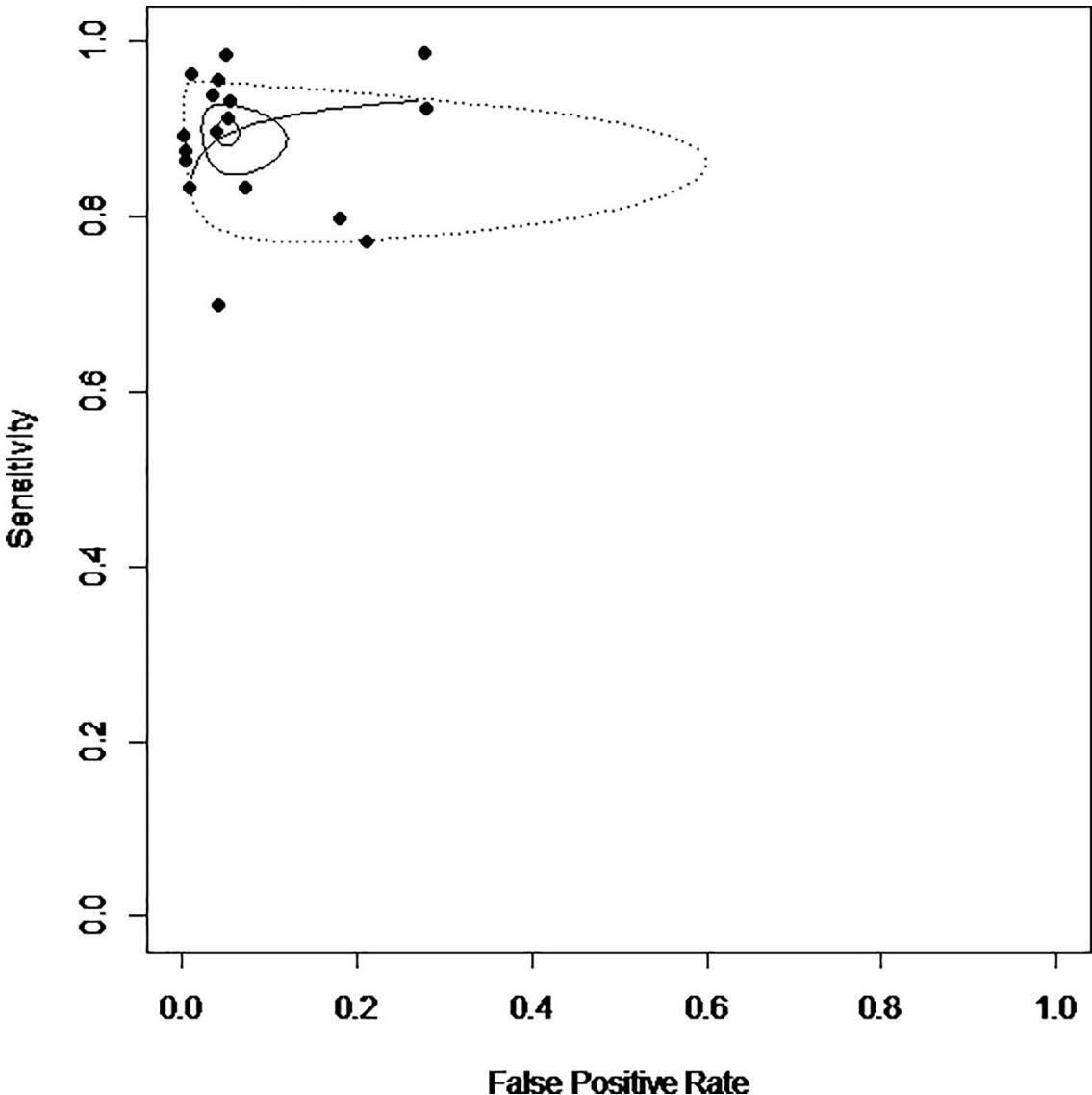

**Fig 3. Summary receiver operating characteristics curve from the hierarchical summary receiver operating characteristic model generated from the eligible studies.** Smaller circle: summary point of sensitivity and false positive rate; sensitivity 0.90 [0.86, 0.92], false positive rate (1-specificity) 0.05 [0.03, 0.1]. Larger circle: 95% confidence region. Dotted circle: 95% prediction region.

$p = 0.6$). Moreover, the meta-regression showed substantial residual heterogeneity ($I^2 =$ 95.7%), indicating that the proportions of BL, LBL, and HCL did not influence the specificity in studies using mouse monoclonal antibodies and that substantial residual heterogeneity in specificity was present after considering the effects of BL, LBL, and HCL.

Third, meta-regression was carried out for studies using rabbit polyclonal antibodies with specified cases of BL, LBL, and HCL.[5, 6, 14, 15, 17] The results showed that the proportions of BL, LBL, and HCL were significant covariates among studies using rabbit polyclonal antibodies (intercept = 1.44, 95% CI, 1.29–1.6, $p < 0.0001$; slope = -1.43, 95% CI, -2.3–0.56, $p = 0.001$). Additionally, meta-regression showed substantial residual heterogeneity ($I^2 =$ 83.8%). However, the residual heterogeneity was relatively lower than the previous two meta-regressions. These results indicated that there was an inverse relationship between the

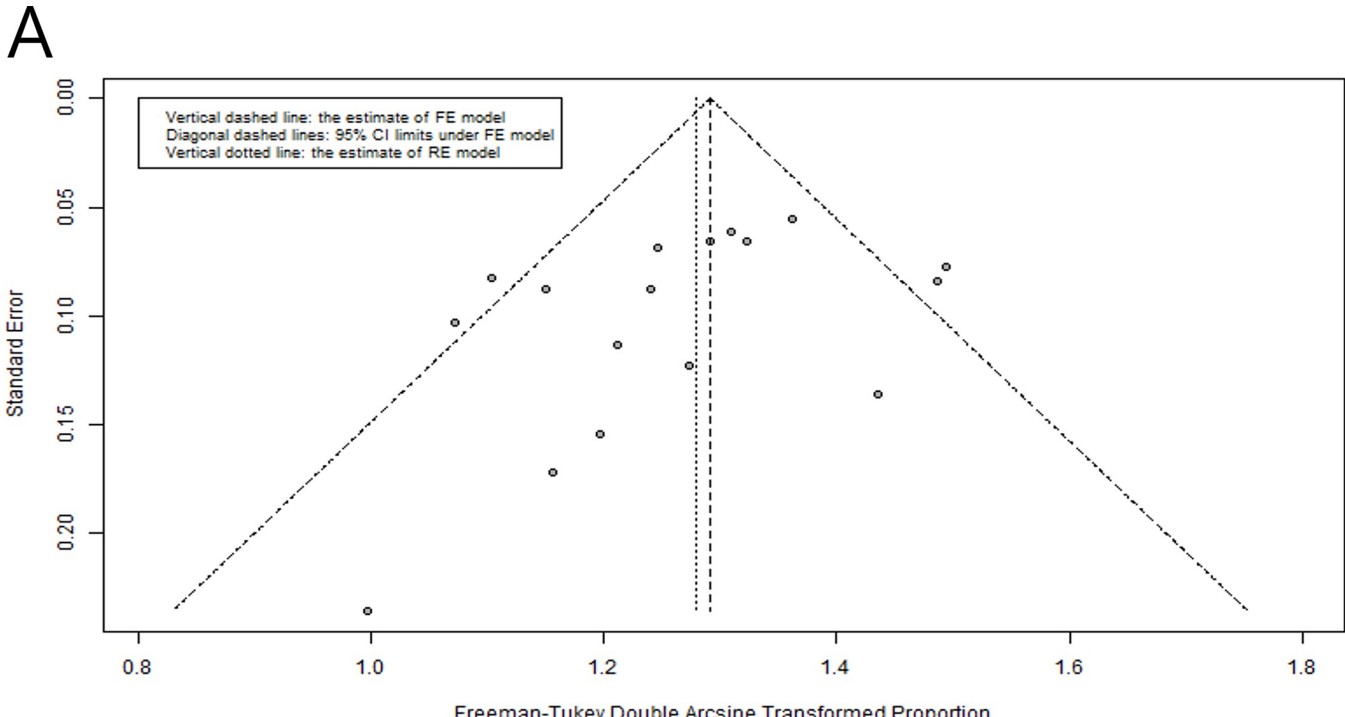

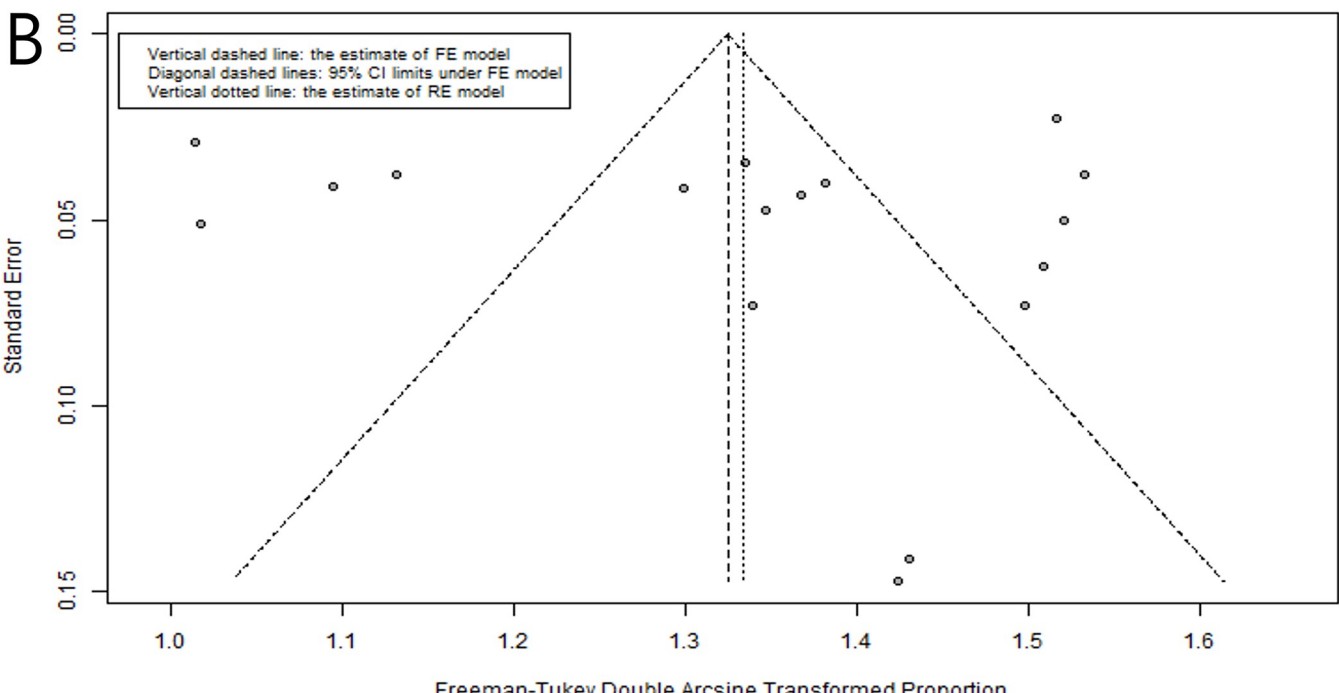

**Fig 4. Funnel plot of meta-analysis.** (A) sensitivity and (B) specificity. FE = fixed effect and RE = random effect.

A

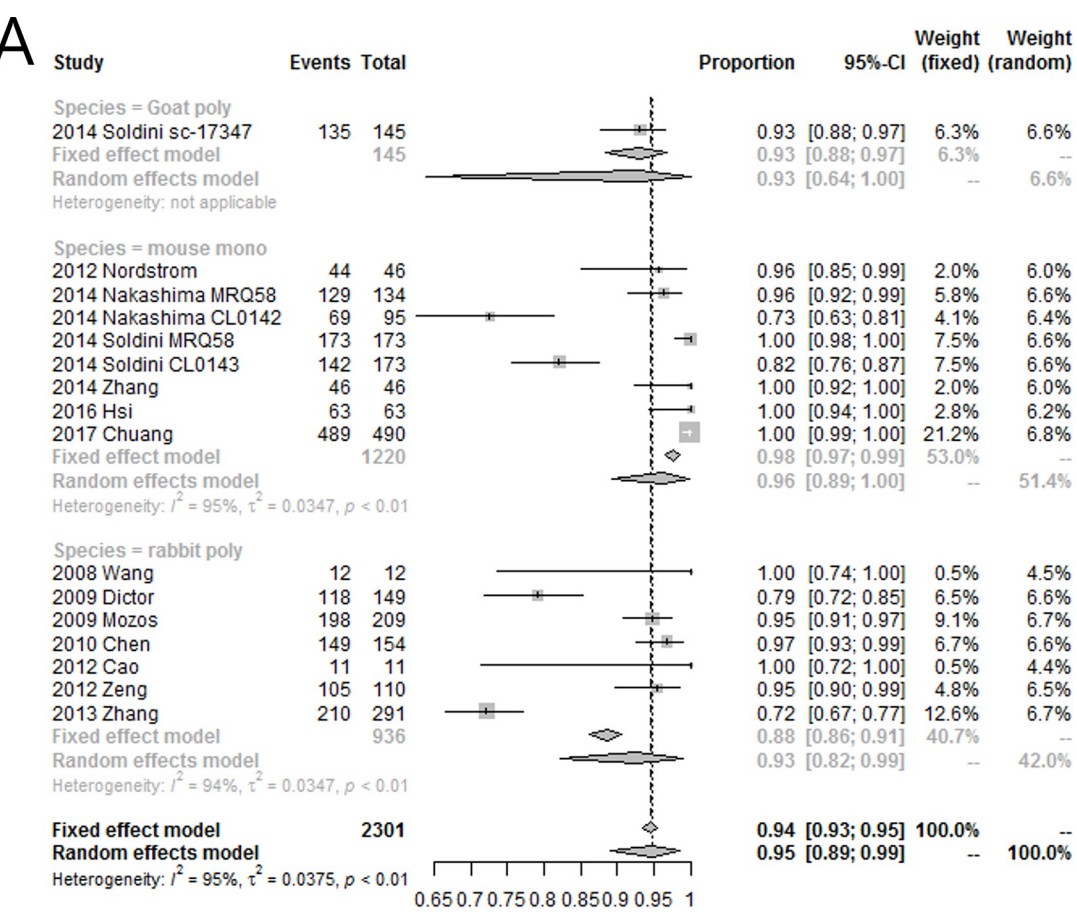

B

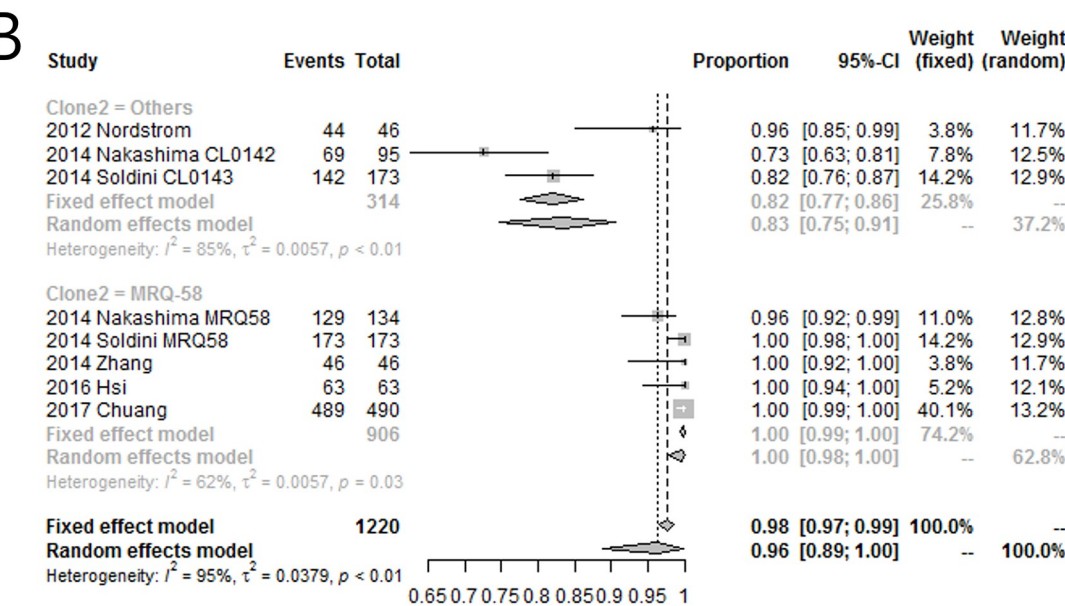

**Fig 5. Forest plot of subgroup analysis.** (A) Subgroup analysis by antibody clonality and (B) clone of monoclonal antibodies.

**Table 2. Meta-regression results.**

| Studies included in meta-regression | Coefficient | | | Intercept | | | $I^2$ (%) |
|---|---|---|---|---|---|---|---|
| | estimate | *P* value | 95% CI | estimate | *P* value | 95% CI | |
| All studies | -0.23 | 0.68 | -1.33 to 0.86 | 1.32 | <0.001 | 1.09 to 1.55 | 94.8 |
| Mouse monoclonal antibody studies | 0.56 | 0.6 | -1.56 to 2.64 | 1.2 | <0.0001 | 0.73 to 1.67 | 95.7 |
| Rabbit polyclonal antibody studies | -1.43 | 0.001 | -2.3 to -0.56 | 1.44 | <0.0001 | 1.29 to 1.6 | 83.8 |

proportions of BL, LBL, and HCL and specificity in studies using rabbit polyclonal antibodies but substantial residual heterogeneity was present in specificity after considering the effects of BL, LBL, and HCL.

## Discussion

In the current meta-analysis, we evaluated the diagnostic accuracy of SOX11 immunohistochemistry for diagnosis of MCL among LPDs. The results demonstrated that SOX11 was a potential diagnostic marker for MCL with a pooled sensitivity and specificity of 0.9 and 0.95, respectively. Heterogeneity of specificity was further explored with subgroup analysis and meta-regression, and meta-regression revealed a significant inverse relationship between specificity and proportions of BL, LBL, and HCL.

The pooled sensitivity was 0.9, and there was no substantial heterogeneity ($I^2$ = 49%). Potential sources of false-negative results are leukemic non-nodal MCL, small cell variant MCL, and aggressive MCL, which tend to show low SOX11 expression.[7, 17, 30, 31] Meta-regression with proportions of such cases could have explained the heterogeneity observed in the sensitivity; however, the regression could not be performed because the included studies did not specify the number of such cases.

The pooled specificity was 0.95, and there was substantial heterogeneity ($I^2$ = 95%). We suspected that the source of the heterogeneity may be the use of less specific polyclonal antibodies. However, the subgroup analysis result between antibody clonality showed statistically significant within- and between-group heterogeneity. Because the specificity of a MRQ-58 clone was thought to be superior to that of other mouse monoclonal antibodies,[12, 21] we performed subgroup analysis between the MRQ-58 clone and the remaining mouse monoclonal antibodies. Although the specificity of four of five studies in this group was 1, the subgroup analysis showed statistically significant substantial heterogeneity in the MRQ-58 group. This result could be explained by the observation that specificities close to 1 have very small standard errors, implying very short confidence intervals. Therefore, when the number of studies in the meta-analysis was low, the specificity of MRQ-58 was considered consistently high in the clinical setting, despite the statistically significant heterogeneity. With meta-regression, we found a statistically significant inverse relationship between specificity and proportions of BL, LBL, and HCL in studies using rabbit polyclonal antibodies. In other words, the specificity of rabbit polyclonal antibodies decreased as the proportions of BL, LBL, and HCL increased. In addition, rabbit polyclonal antibodies showed higher false-positive rates for diffuse large B cell lymphoma (DLBCL) as compared with mouse monoclonal antibodies, particularly for the MRQ-58 clone.[5, 19, 21]

With sensitivity and specificity from eligible studies, we plotted SROC curves and obtained an AUC of 0.934, indicating that SOX11 had good accuracy in the diagnosis of MCL from other LPDs. SOX11 is a good diagnostic marker for MCL, particularly in the diagnosis of cyclin D1-negative MCL and to distinguish between aggressive MCL and CD5-positive DLBCL.[7] However, because a small subset of MCL is SOX11-negative, the diagnostic

applications of SOX11 should be incorporated in an immunohistochemistry panel approach rather than used alone.

This meta-analysis was limited by the relatively small number of studies included. In particular, a meta-regression of sensitivity, using possible factors causing false positives as covariates, could not be performed because the articles did not record such data. Additionally, residual heterogeneity of meta-regression in specificity could not be explored further because of a lack of additional data. Thus, we speculate that the possible source of residual heterogeneity could be different staining protocols, varying specimen conditions, and different cut-off thresholds for immunohistochemistry interpretation.

A previous study showed that differences in cut-off values used for the interpretation of SOX11 immunohistochemistry data between studies was an important source of inconsistency in results.[32] For the definition of negative SOX11 staining, 8 among the 14 studies included in this analysis specified a cut-off for SOX11 immunohistochemistry. 5 of these studies used 10% as the cut-off value, two used 20%, and one used 1%. We also think differences in cut-off values could affect study results, but additional detailed analysis in this direction is beyond the scope of this paper.

In conclusion, the current evidence suggested that SOX11 may be a useful diagnostic immunohistochemical marker for MCL. In particular, clone MRQ-58 mouse monoclonal antibody showed robust performance. Future studies using MRQ-58 are needed to improve our understanding of the diagnostic accuracy of SOX11 immunohistochemistry for MCL.

## Supporting information

**S1 Table. QUADAS evaluation.** QUADAS questions and answers are listed.
(XLSX)

**S2 Table. PRISMA 2009 checklist.**
(DOC)

## Author Contributions

**Conceptualization:** Hyunchul Kim.

**Data curation:** Woojoo Lee.

**Formal analysis:** Woojoo Lee.

**Investigation:** Eun Shin, Bo-Hyung Kim, Hyunchul Kim.

**Project administration:** Eun Shin, Bo-Hyung Kim.

**Supervision:** Hyunchul Kim.

**Writing – original draft:** Hyunchul Kim.

**Writing – review & editing:** Woojoo Lee, Eun Shin, Bo-Hyung Kim, Hyunchul Kim.

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
