## [Decision Letter · Decision Letter 0]

5 Sep 2019

PONE-D-19-19834

Diagnostic accuracy of SOX11 immunohistochemistry in mantle cell lymphoma: a meta-analysis

PLOS ONE

Dear Dr. Kim,

Thank you for submitting your manuscript to PLOS ONE. After careful consideration, we feel that it has merit but does not fully meet PLOS ONE’s publication criteria as it currently stands. Therefore, we invite you to submit a revised version of the manuscript that addresses the points raised during the review process.

In particular, the study need a more careful statistical assessment as pointed out by reviewers. The definition of SOX11 negativity should be clarified as well as the criteria used to identify SOX11-negative cases. Differences among different antigen retrievals I methods in the various studies should be also considered. In addition, the Authors should provide a more convincing description of the novelty and significance of the data obtained in the light of their potential relevance in the diagnosis of MCL.

We would appreciate receiving your revised manuscript by Oct 20 2019 11:59PM. To enhance the reproducibility of your results, we recommend that if applicable you deposit your laboratory protocols in protocols.io, where a protocol can be assigned its own identifier (DOI) such that it can be cited independently in the future. For instructions see: http://journals.plos.org/plosone/s/submission-guidelines#loc-laboratory-protocols

We look forward to receiving your revised manuscript.

Kind regards,

Riccardo Dolcetti

Academic Editor

PLOS ONE

Journal Requirements:

2. Please upload a new copy of Figures 2A, 2B, 5A and 5B as the detail is not clear. Please follow the link for more information: http://blogs.PLOS.org/everyone/2011/05/10/how-to-check-your-manuscript-image-quality-in-editorial-manager/

3. Please provide any updates you might have since the original search was performed in May 2018, or please provide the rational for ending your search at that time.

Reviewers' comments:

Reviewer's Responses to Questions

**Comments to the Author**

1. Is the manuscript technically sound, and do the data support the conclusions?

Reviewer #1: Yes

Reviewer #2: Yes

Reviewer #3: Partly

2. Has the statistical analysis been performed appropriately and rigorously? 

Reviewer #1: I Don't Know

Reviewer #2: Yes

Reviewer #3: No

3. Have the authors made all data underlying the findings in their manuscript fully available?

Reviewer #1: Yes

Reviewer #2: Yes

Reviewer #3: Yes

4. Is the manuscript presented in an intelligible fashion and written in standard English?

Reviewer #1: Yes

Reviewer #2: Yes

Reviewer #3: Yes

5. Review Comments to the Author

Reviewer #1: The work reports the results of a meta-analysis focused on the diagnostic accuracy of the immunohistochemical analysis of SOX11 in mantle cell lymphoma.

SOX11 is a tumor marker particularly useful in the recognition of CydinD1 negative mantle cell lymphoma. Since SOX11 is "falsely" positive in cases of Burkitt lymphoma, lymphoblastic lymphoma and Hairy cell leukemia, the authors investigated the value of SOX11 against other lymphoproliferative disorders in which it is expressed.

The meta-analysis study was therefore limited to 14 of the 383 reports collected. From these 14 studies it was found that SOX 11, evaluated with the MRQ-58 mouse monoclonal antibody can be a useful diagnostic marker by immunohistochemistry, also in statistical comparison with the other examined lymphomas that showed false positivity for SOX11.

Reviewer #2: In this article, the authors performed extensive literature search to evaluate the accuracy of SOX11 immunostain in the diagnosis of mantle cell lymphoma using the meta-analysis on the published data. They concluded that SOX11 is a useful marker in diagnosing MCL, and the SOX11 mouse monoclonal antibody clone, MRQ-58, has robust performance in contrast to other antibodies. Overall, the paper is well-written, and the results are reliable. However, the findings are not novel or significant, and are of limited usage in the diagnosis of MCL.

Specific comments:

1). Was there any specific definition(s) of “negative SOX11 staining” in the literature?

2). During immunohistochemical staining, the antigen retrieval methods may significantly affect the quality of results. Therefore, it may be helpful for the authors to analyze the differences among different antigen retrievals of SOX11 immunostain.

3). For the SOX11-negative MCLs, how was the diagnosis established?

Reviewer #3: This manuscript reports the summarized results of using SOX11 in diagnostic test of MCL. I have below questions and comments.

Please add a table to show the results of quality evaluation of QUADAS.

Lines 114-117, threshold effect was mentioned. Were various thresholds used among cited studies? If yes, please report the cut-off value of each study. The Spearman r=0.99, very high, why no threshold effect was concluded? What are your interpretations for logPLR, logNLR and DOR in this meta-analysis results?

Fig 3. Line 124, please add “summary point of” before “sensitivity and false positive rate”. It indicates HSROC, but SROC is mentioned at lines 110-111.

For Figure 4, please provide legends for each line.

6. PLOS authors have the option to publish the peer review history of their article (what does this mean?). If published, this will include your full peer review and any attached files.

Reviewer #1: Yes: Annunziata Gloghini

Reviewer #2: No

Reviewer #3: No

---

## [Author Response · Author response to Decision Letter 0]

13 Oct 2019

Dear Dr. Dolcetti,

Thank you for giving us the opportunity to submit a revised draft of our manuscript titled “Diagnostic accuracy of SOX11 immunohistochemistry in mantle cell lymphoma: a meta-analysis” to PLOS ONE. We appreciate the time and effort that you and the reviewers have dedicated to providing your valuable feedback on our manuscript. We are grateful to the reviewers for their insightful comments on our paper. We have been able to incorporate changes to reflect many of the suggestions provided by the reviewers. We have highlighted the changes within the manuscript.

Here is a point-by-point response to the reviewers’ comments and concerns.

Academic Editor

1. Please ensure that your manuscript meets PLOS ONE's style requirements, including those for file naming. The PLOS ONE style templates can be found at http://www.journals.plos.org/plosone/s/file?id=wjVg/PLOSOne_formatting_sample_main_body.pdf and http://www.journals.plos.org/plosone/s/file? id=ba62/PLOSOne_formatting_sample_title_authors_affiliations.pdf

Response: We have ensured that the revised manuscript conforms with the style requirements of PLOS ONE.

2. Please upload a new copy of Figures 2A, 2B, 5A and 5B as the detail is not clear. Please follow the link for more information: http://blogs.PLOS.org/everyone/2011/05/10/how-to-check-your-manuscript-image-quality-in-editorial-manager/

Response: Figures 2A, 2B, 5A and 5B are newly created to show the detail clearly. 

3. Please provide any updates you might have since the original search was performed in May 2018, or please provide the rational for ending your search at that time.

Response: You have raised an important point here. And we also had considered renewal of searched articles, but we chose not to. There are several reasons for that.

First, this meta-analysis required several months to be completed. If search was renewed, the whole process would have to be done all over again and similar amount of time would have to be spent. So the search had to be stopped somewhere and it was May 2018.

Second, there was no additional article with available data searched until recently. Even after we decided to stop searching on May 2018, we searched databases regularly to see if there were new critical articles to be added. But until recently there was no new article with available data published. We searched database recently (September 6th, 2019), and found three new relevant articles below.

1. SOX11- negative Mantle Cell Lymphoma: Clinicopathologic and Prognostic Features of 75 Patients. Xu J, Am J Surg Pathol. 2019 May;43(5):710-716. doi: 10.1097/PAS.0000000000001233., 

2. CD23 expression in mantle cell lymphoma is associated with CD200 expression, leukemic non-nodal form, and a better prognosis. Saksena A, Hum Pathol. 2019 Jul;89:71-80. doi: 10.1016/j.humpath.2019.04.010. Epub 2019 May 2.,

3. Highly sensitive and specific in situ hybridization assay for quantification of SOX11 mRNA in mantle cell lymphoma reveals association of TP53 mutations with negative and low SOX11 expression. Federmann B, Haematologica. 2019 Jul 11. pii: haematol.2019.219543. doi: 10.3324/haematol.2019.219543. [Epub ahead of print])

As you can see, May 2019 is the earliest published date. The rest are published on July 2019. The published dates were too late for the data to be added. And we were unaware of the articles because by the time our manuscript was nearly completed and we could not renew the search.

Response: A supporting Information file is submitted and caption was newly inserted at the end of manuscript.

Reviewer #1: The work reports the results of a meta-analysis focused on the diagnostic accuracy of the immunohistochemical analysis of SOX11 in mantle cell lymphoma. SOX11 is a tumor marker particularly useful in the recognition of CydinD1 negative mantle cell lymphoma. Since SOX11 is "falsely" positive in cases of Burkitt lymphoma, lymphoblastic lymphoma and Hairy cell leukemia, the authors investigated the value of SOX11 against other lymphoproliferative disorders in which it is expressed. The meta-analysis study was therefore limited to 14 of the 383 reports collected. From these 14 studies it was found that SOX 11, evaluated with the MRQ-58 mouse monoclonal antibody can be a useful diagnostic marker by immunohistochemistry, also in statistical comparison with the other examined lymphomas that showed false positivity for SOX11.

Reviewer #2: In this article, the authors performed extensive literature search to evaluate the accuracy of SOX11 immunostain in the diagnosis of mantle cell lymphoma using the meta-analysis on the published data. They concluded that SOX11 is a useful marker in diagnosing MCL, and the SOX11 mouse monoclonal antibody clone, MRQ-58, has robust performance in contrast to other antibodies. Overall, the paper is well-written, and the results are reliable. However, the findings are not novel or significant, and are of limited usage in the diagnosis of MCL.

Specific comments:

1). Was there any specific definition(s) of “negative SOX11 staining” in the literature?

Response: Thank you for pointing this out. Some papers defined “negative SOX11 staining” with specific cut-off value. But others did not specify the definition. We revised the manuscript to add discussion on the issue. Regarding the interpretation of SOX11 immunohistochemistry, we briefly discussed the definition of “negative SOX11 staining” and cited a previous paper about effects of SOX11 staining interpretation on inconsistency between studies.

2). During immunohistochemical staining, the antigen retrieval methods may significantly affect the quality of results. Therefore, it may be helpful for the authors to analyze the differences among different antigen retrievals of SOX11 immunostain.

Response: We agree with your opinion on significance of antigen retrieval methods and wish we could perform subgroup analysis according to the antigen retrieval methods. But as you can see below, antigen retrieval methods used for each article are highly heterogeneous and such an analysis seems unavailable.

Detailed explanation is as follows. Most of the articles stated that they used heat-induced epitope retrieval. Two articles did not specify their antigen retrieval methods. Four studies used automated antigen retrieval methods (three with PT-LINK from Dako and two with BondMax from Leica). The rest are heat-induced epitope retrieval methods with varied equipment, reagents, and settings. To perform subgroup analysis, the studies needs to be divided into two or more groups. The best possible grouping seems to be automated group VS heat-induced group. But each group is highly heterogeneous; Automated group is composed of three PT-LINK used studies with different settings and two BondMax used studies with different settings. And heat-induced group is composed of all different combinations of equipment, reagents, and settings. Therefore subgroup analysis with simple two groups of automated group VS heat-induced group would be inappropriate. If, instead, studies are divided into subgroups according to individual combinations of equipments, reagents, the analysis would produce more than 10 subgroups with mostly one study per subgroup and the result would be meaningless.

In short, we agree with your opinion and we would much like to perform additional subgroup analysis. But because of the above mentioned limitations, proper analysis does not seem available. 

Antigen Retrieval Methods of Articles

2008 Wang Heat-induced (Retriever 2100)

2009 Dictor Heat-induced (Microwave, Tris/EDTA, pH9, 8+7min)

2009 Mozos ER2, BondMax, Leica, 15min

2010 Chen Heat-induced (Decloaking Chamber, Biocare Medical, pH6.0)

2012 Cao Heat-induced (Decloaking Chamber, Biocare Medical, pH6.0)

2012 Hsiao N/A

2012 Nordstrom PT-LINK, Dako, pH9

2012 Zeng Heat-induced (Tris-based buffer, pH9.0)

2013 Zhang N/A

2014 Nakashima MRQ58 Heat-induced (EDTA, pH8.9-9.1, 20min)

2014 Nakashima CL0142 Heat-induced (EDTA, pH8.9-9.1, 20min)

2014 Soldini MRQ58 PT-LINK, low, Dako, 24 degree, 30 min

2014 Soldini CL0143 PT-LINK, high PH, Dako, 24 degree, 60 min

2014 Soldini sc-17347 ER1, high PH, BondMax, Leica, 30 degree, 60 min

2014 Zhang Heat-induced (citrate buffer, pH 6.0, 20min)

2016 Hsi Heat-induced (Ultra Cell Conditioning Solution, 95 degree for 65 min)

2017 Chuang Heat-induced (EDTA, 100 degree, 30 min)

3). For the SOX11-negative MCLs, how was the diagnosis established?

Response: We would like to answer the question in two parts.

First, we would like to answer first part of the question regarding SOX11-negative immunohistochemistry. As stated in answer for question No.1, some authors used specific cut-off value to interpret SOX11 immunohistochemistry. Others did not specify how they interpreted the immunohistochemistry.

The second part of the question seems to be about establishing diagnosis of MCL with SOX11 negative immunohistochemistry. The authors’ diagnoses were in accordance with well-known definition of MCL (such as small to intermediate cell size, and CD5 and Cyclin D1 co-expression).

Reviewer #3: This manuscript reports the summarized results of using SOX11 in diagnostic test of MCL. I have below questions and comments.

Please add a table to show the results of quality evaluation of QUADAS.

Response: The QUADAS questions and answers are provided as a Supplement Material (S1 Table). QUADAS scores are added in the far right column of Table.1.

Lines 114-117, threshold effect was mentioned. Were various thresholds used among cited studies? If yes, please report the cut-off value of each study. The Spearman r=0.99, very high, why no threshold effect was concluded? What are your interpretations for logPLR, logNLR and DOR in this meta-analysis results?

Response: Thank you for pointing out our mistake. The Spearman correlation was not 0.99, but 0.099. We corrected the number. A good diagnostic test should have large log PLR, small log NLR, and large DOR. The overall effect sizes of log PLR and log NLR, which are significantly away from 0, imply that SOX11 immunohistochemistry has diagnostic value for MCL. Following McGee[28], the overall effect size of DOR also shows clear evidence of SOX11 immunohistochemistry as a diagnostic test for MCL. This interpretation is added in Diagnostic accuracy of SOX11 for MCL.

Fig 3. Line 124, please add “summary point of” before “sensitivity and false positive rate”. It indicates HSROC, but SROC is mentioned at lines 110-111.

Response: The words and phrases are added at the positions.

For Figure 4, please provide legends for each line.

Response: We added legends for each line. See Figure 4. The vertical dashed line corresponds to the estimate of the fixed effect model, the diagonal dashed lines represent 95% confidence interval limits under the fixed effect model, and the vertical dotted line corresponds to the estimate of the random effect model.

We look forward to hearing from you in due time regarding our submission and to respond to any further questions and comments you may have.

Sincerely,

Hyunchul Kim

---

## [Editor Report · Decision Letter 1]

30 Oct 2019

Diagnostic accuracy of SOX11 immunohistochemistry in mantle cell lymphoma: a meta-analysis

PONE-D-19-19834R1

Dear Dr. Kim,

We are pleased to inform you that your manuscript has been judged scientifically suitable for publication and will be formally accepted for publication once it complies with all outstanding technical requirements.

With kind regards,

Riccardo Dolcetti

Academic Editor

PLOS ONE
---

## [Editor Report · Acceptance letter]

5 Nov 2019

PONE-D-19-19834R1 

Diagnostic accuracy of SOX11 immunohistochemistry in mantle cell lymphoma: a meta-analysis 

Dear Dr. Kim:

I am pleased to inform you that your manuscript has been deemed suitable for publication in PLOS ONE. Congratulations! Your manuscript is now with our production department. 

With kind regards,

on behalf of

Dr. Riccardo Dolcetti 

Academic Editor

PLOS ONE